# Targeting the Endocannabinoidome in Pancreatic Cancer

**DOI:** 10.3390/biom12020320

**Published:** 2022-02-17

**Authors:** Valerio Falasca, Marco Falasca

**Affiliations:** 1School of Chemistry, The University of New South Wales, Sydney, NSW 2052, Australia; V.falasca@unsw.edu.au; 2Metabolic Signalling Group, Curtin Health Innovation Research Institute, Curtin Medical School, Curtin University, Perth, WA 6102, Australia

**Keywords:** endocannabinoid system, pancreatic ductal adenocarcinoma, endocannabinoids, G protein-coupled receptors, cannabinoid receptors

## Abstract

Pancreatic Ductal adenocarcinoma (PDAC), the most common malignancy of the pancreas, is an aggressive and lethal form of cancer with a very high mortality rate. High heterogeneity, asymptomatic initial stages and a lack of specific diagnostic markers result in an end-stage diagnosis when the tumour has locally advanced or metastasised. PDAC is resistant to most of the available chemotherapy and radiation therapy treatments, making surgery the most potent curative treatment. The desmoplastic tumour microenvironment contributes to determining PDAC pathophysiology, immune response and therapeutic efficacy. The existing therapeutic approaches such as FDA-approved chemotherapeutics, gemcitabine, abraxane and folfirinox, prolong survival marginally and are accompanied by adverse effects. Several studies suggest the role of cannabinoids as anti-cancer agents. Cannabinoid receptors are known to be expressed in pancreatic cells, with a higher expression reported in pancreatic cancer patients. Therefore, pharmacological targeting of the endocannabinoid system might offer therapeutic benefits in pancreatic cancer. In addition, emerging data suggest that cannabinoids in combination with chemotherapy can increase survival in transgenic pancreatic cancer murine models. This review provides an overview of the regulation of the expanded endocannabinoid system, or endocannabinoidome, in PDAC and will explore the potential of targeting this system for novel anticancer approaches.

## 1. Introduction

Cannabis has been used for millennia both as a recreational drug and for its medicinal properties [1]. As early as 2737 B.C., cannabis was prescribed by Chinese emperor Shen Nung to treat a variety of ailments [2]. Recently, marijuana use has spread to become the most used illicit drug worldwide, being recently legalised in several countries [3]. Research into the identification of the chemical components of the *Cannabis Sativa* plant continued for over a century, but due to limitations of analytical equipment in the early 19th century, its main psychoactive constituent, Δ^9^-tetrahydrocannabidiol (THC), was not synthesised and characterised in its pure enantiomeric form until 1984 [4,5]. Over time, more than 120 cannabis-derived cannabinoids have been identified and collectively called “phytocannabinoids” [6]. In addition, a group of endogenous lipids that mimic the activity of THC and are involved in numerous physiological and pathological functions have gradually been discovered and studied, and grouped as “endocannabinoids” for their effects similar to those caused by phytocannabinoids [7]. These findings, and their potential therapeutical exploitation, promoted increasing interest among pharmaceutical companies to produce compounds chemically akin to phyto- and endocannabinoids, namely “synthetic cannabinoids” [8].

Due to its multiple clinical effects including euphoria, appetite stimulation and immune modulation, THC was studied in-depth to determine its mechanism of action [9]. Consequently, the identification of THC’s molecular targets in humans led to the discovery of two G-protein-coupled receptors (GPCRs), namely CB_1_ and CB_2_, which were named cannabinoid receptors [10]. Subsequent studies enabled the detection of endogenous ligands (endocannabinoids) for these previously orphan receptors, as well as five enzymes involved in their biosynthesis and catabolism [11]. These initial discoveries gave rise to the endocannabinoid system (ECS), a term used to describe the newly discovered endogenous lipid signalling system, which was associated with many pathologies including cancer, chronic pain, inflammation, epilepsy, anxiety, multiple sclerosis, Huntington’s disease and more [12]. At the turn of the century however, it became increasingly evident that the cannabinoid signalling system is more complex than what had initially been idealised. Indeed, hundreds of related cannabinoid ligands, receptors and biosynthetic and catabolic enzymes are thought to take part in ECS signalling, which calls for the use of the more comprehensive term endocannabinoidome (eCBome), to encompass the expanded view of this system [13]. The eCBome plays a role in a wider range of physiological roles than once thought, as well as being involved in a greater number of diseases. Among these diseases is pancreatic cancer, one of the most aggressive and complicated malignancies to treat, which demonstrates altered expressions of cannabinoid receptors and molecules [14]. Due to the common late diagnosis and aggressive metastatic spread of this disease, its prognosis has hardly improved from a discouraging 5-year survival rate below 10% [15]. Currently, some cannabinoid drugs, e.g. nabilone, are approved for clinical use; however, the principal application is for palliative care [16]. There is evidence to suggest cannabinoids can exert anti-cancer effects, especially in combination with existing therapies, although this has not yet been fully proven, as the majority of this research has been conducted in vitro or in animal models, which cannot fully gauge the action of the human eCBome system [17]. The use of medicinal cannabinoids against cancer is still partly controversial, with a portion of studies suggesting they may reduce response to treatment or increase tumour progression in some cancers. What is known is that the eCBome plays a functional role in pancreatic cancer regulation, and understanding its signalling mechanisms offers the potential for new or improved treatments.

## 2. Endocannabinoidome

The initial discovery of the endocannabinoid system was a major milestone in the understanding of the value and mechanisms of medicinal cannabinoids. This novel system, uncovered from the molecular actions of THC, consisted of (1) CB_1_ and CB_2_ receptors [10]; (2) their two main endogenous receptor agonists, *N*-arachidonoylethanolamine (AEA) and 2-arachidonoyl-glycerol (2-AG), as well as their synthetic precursors [18,19]; (3) enzymes catalysing the biosynthesis of the endocannabinoids, namely N-acyl-phosphatidylethanolamine-specific phospholipase D-like (NAPE-PLD) and the sn-1 selective diacylglycerol lipases α and β (DAGLα and DAGLβ) [20,21,22]; (4) hydrolase enzymes which non-selectively degrade anandamide (AEA) and 2-arachidonoylglycerol (2-AG), respectively, fatty acid amide hydrolase (FAAH) and monoacylglycerol lipase (MAGL) [23,24]. The ligands AEA and 2-AG, belonging to the N-acylethanolamine (NAE) and monoacylglycerol (MAG) families, respectively, were classified as cannabinoids and gained a significant amount of research attention (Figure 1). In fact, increasing evidence demonstrated the involvement of the ECS in a wide range of physiological roles in the neural, cardiovascular, digestive and immune systems, offering pharmacological opportunities beyond the scope of THC [25].

Since the discovery of the ECS and its potential benefits in disease treatment, a significant amount of attention was invested in understanding the roles of ECS signalling in the body. Over time, various research in cannabinoids’ mechanisms of action, synthesis and catabolism provided evidence to suggest that there are many additional cannabinoid analogues, receptors and enzymes associated with this system (Figure 2). Indeed, the agonists AEA and 2-AG showed an affinity for receptors other than CB_1_ and CB_2_ [26,27,28], including the G protein-coupled receptor 55 (GPR55) [29], the transient receptor potential vanilloid subtype 1 (TRPV1) [30], and the peroxisome proliferator-activated receptors (PPARs) [31].

The cannabinoid-binding receptor GPR55 is largely present in the brain, skeletal muscle, gastrointestinal (GI) tract, pancreas, white adipose tissue, β and α cells of the islets of Langerhans, and is studied for its role in maintaining glucose and energy balance, as well as cancer-promoting activity [32,33]. The GPR119 receptor, expressed in the pancreas and GI tract, has gained importance in the fight against type 2 diabetes and obesity as an insulin and glucose regulator [34]. GPR18 has been found especially in immune cells, as well as in the brain, GI tract and white adipose tissue; despite its low affinity with CB1 and CB2 receptors, it is sensitive to many cannabinoids and it is thought to be involved in a number of biological activities, in particular anti-inflammatory responses [35]. The capsaicin receptor or TRPV1 is another new member of the ECS extended family; expressed centrally and peripherally in the body, it is correlated to thermogenesis and pain regulation [36] and seems to play a part in obesity and T2D [37]. Similarly, nuclear receptors PPARs are activated by endocannabinoids and endocannabinoid-like molecules. PPARs are vastly expressed in tissues throughout the body and are divided into three isoforms, α, δ and γ. PPARs are key players in biological functions such as cellular specialization, metabolism, development as well as carcinogenesis [38]. In turn, these orphan receptors were found to have other ligands other than AEA and 2-AG modulating their activity, also belonging to the two main families of *N*-acylamides and monoacylglycerols (Figure 1). For example, lysophosphatidylinositol (LPI) was identified as the endogenous ligand for GPR55 [39]. Furthermore, other AEA and 2-AG analogues, including other MAGs and NAE’s such as palmitoylethanolamine (PEA) and oleoylethanolamide (OEA) [40], and *N*-acyl amino acids [41] were found to have similar or shared biosynthetic and metabolic pathways. These numerous congeners effectuate poor or no activity on CB_1_ and CB_2_ receptors; however, they were shown to bind other non-cannabinoid receptors. [28,42,43]. Thus, the evidence suggests there are more factors involved in the ECS signalling than was initially thought, due to the promiscuity of ligands and redundancy in their biological pathways. Hence, the term “endocannabinoidome” was devised in order to classify these novel endogenous lipid mediators, enzymes and receptors, as a broader “ome”, meaning a completed set [44]. Altogether, the eCBome system is greatly expressed across the body, including the CNS, adipose and inflammatory tissue, the GI tract and the pancreas [45]. Therefore, understanding the eCBome poses a significant value for the pharmacological treatment of a multitude of pathological conditions including neuropsychiatric diseases such as anxiety and depression, neurological disorders such as multiple sclerosis, nausea induced by chemotherapy, inflammatory and autoimmune diseases, obesity and type 2 diabetes, and cancers, such as pancreatic cancer.

## 3. Pancreatic Cancer

Pancreatic cancer is among the most challenging malignancies to treat, with one of the highest mortality rates out of all cancers. Pancreatic ductal adenocarcinoma (PDAC) accounts for 90% of pancreatic cancers. Despite significant advancements in the surgical and therapeutic treatment of this disease, there has been little improvement in mortality over the past few decades [15]. Currently, the 5-year survival rate for pancreatic cancer patients amounts to less than 10%, which is one of the lowest out of all cancers [46,47]. Due to the severe pain and cachexia associated with this disease, especially in the later stages, even implementing adequate palliative care has proven to be problematic [48,49].

The poor results in the improvement of this prognosis are mainly due to the late detection of PDAC, increasing the likelihood of metastases and the aggressive spread of this malignancy across the body, rendering it impervious to treatment. Detection of PDAC is impaired by a lack of visible symptoms and accurate diagnostic biomarkers and, as a result, 80% of diagnosed patients are already at an advanced or metastatic stage, against which current treatments are ineffective [50]. As the disease progresses, genetic variation has been observed to alter signalling pathways, leading to an increase in the virulence of each stage. In fact, PDAC often results in the overexpression of many signalling pathways involved in growth and proliferation along with a reduction of tumour suppressor genes, which gift the disease its hostile ability to survive and spread, even against treatments such as chemotherapy, radiotherapy and targeted therapy. Furthermore, PDAC is characterised by a dense, collagenous stromal reaction around its tumour structures, which further contributes to treatment resistance, dysregulated proliferation and invasion [51,52]. This stromal reaction around pancreatic tumour cells comprises the tumour microenvironment (TME), which consists of pancreatic stromal cells (PSCs), extracellular matrix, neural and immune cells, and blood vessels. PSCs interact with pancreatic cancer cells and migrate to compromised sites to exacerbate tumour growth, invasion, immunosuppression and chemoresistance [53].

The first-line treatment for PDAC has been chemotherapy with the antimetabolite gemcitabine since 1997 [54]. However, even when used in combination with other anti-cancer agents, therapy has shown only minimal improvements in survival, at the cost of substantial toxicity and side effects [55]. The only potentially curative treatment for patients presenting with PDAC is surgical resection accompanied by adjuvant chemotherapy [56]. Unfortunately, only 15–20% of patients are diagnosed in time for surgery and up to 75% of patients undergoing surgery experience cancer recurrence. For patients with metastatic disease, gemcitabine plus Nab-paclitaxel (paclitaxel bound to human albumin) is the usual initial regimen [57]. Another combination drug, FOLFIRINOX (5-Fluorouracil, irinotecan, leucovorin, oxaliplatin) is an alternative regimen. Both are associated with nausea, vomiting, poor appetite, lethargy neuropathy and neutropenia. Despite several clinical trials on therapeutic efficacy, only a marginal improvement in patients’ survival has been achieved. The failure to translate the clinical benefit to all PDAC cases is due to the rapid development of a chemoresistance mechanism in patients [58]. Tumour plasticity and heterogeneity, tumour microenvironment, epithelial to mesenchymal transition, nucleoside transporters and their enzymes and transcription factors facilitate chemoresistance to chemotherapeutic agents [58]. With the frequency of incidence emulating mortality, there is an urgent need to develop new and more efficient therapeutic options for PDAC patients while also reducing the burden of side effects.

The USA National Academies of Sciences, Engineering and Medicine note in their exhaustive 2017 review, “The Health Effects of Cannabis and Cannabinoids: The current state of evidence and recommendations for research”, that “there is evidence to suggest that cannabinoids… may play a role in the cancer regulation processes... Therefore, there is interest in determining the efficacy of cannabis or cannabinoids for the treatment of cancer” (p. 90) [59].

Thus, there is a pressing need for innovation in pancreatic cancer treatments, which could be derived from understanding the role of the eCBome system in this disease.

## 4. Expression of Endocannabinoidome Members in Pancreatic Cancer

Research into the involvement of the eCBome on PDAC is hoped to provide new options to tackle this aggressive disease. Interestingly, evidence indicates that eCBome expression and signalling is altered in pancreatic cancer, suggesting a potential relationship between the two.

Studies of patients with PDAC have shown overexpression of both CB_1_ and CB_2_ receptors on pancreatic cells, compared to very limited expression in healthy pancreatic cells [49]. Furthermore, elevated levels of the eCBome metabolising enzymes FAAH and MAGL were also identified in cells of pancreatic cancer patients [49]. This study determined that activation of the overexpressed CB_2_ receptors in particular induced pancreatic cell apoptosis. In addition, the study found that low expression of CB_1_ receptors or high levels of FAAH or MAGL were correlated to longer patient survival [49]. Similarly to the cannabinoid receptors, immunohistochemistry analysis demonstrated diffused expression of GPR55 receptors in PDAC patients. While GPR55 receptors in the pancreas of healthy patients are only present in the Islets of Langerhans, PDAC cells demonstrated a diffused expression of GPR55 [60]. Accordingly, the endogenous ligand for GPR55 receptors, LPI, also shows altered levels in cancers, which are a possible cause for the accumulation of GPR55 [61]. Similarly, GPR119 has been found overexpressed in pancreatic cancer [62] and in silico analysis of the open-access databases indicate that GPR119 is a favourable prognostic factor (Human Protein Atlas) [62]. TRPV1 has also been shown to be expressed in pancreatic cancer cells and tissues and could contribute to inflammation, pain and PDAC progression [63,64]. TRPV1 has been also shown to stimulate epidermal growth factor receptor (EGFR) ubiquitination and regulate its signalling pathways in PDAC cells [64]. In addition, PPARs signalling pathway has been shown to be involved in PDAC development and progression [65].

Pancreatic cancer has a significantly elevated immune resistance compared to most cancers. Nonetheless, PDAC tumours recruit various immune cells, which express CB_1_ and CB_2_ cannabinoid receptors. These immune cells include T cells, B cells, natural killer cells, dendritic cells, macrophages, eosinophils, neutrophils, monocytes and mast cells. This significant expression suggests that cannabinoids play a regulatory role in cancer’s immune system, affording the name ‘immune endocannabinoid system’. Research on the concentrations of eCBome ligands related to the tumour-immune microenvironment showed differing trends. It was observed that 2-AG is secreted by tumours, and its concentrations are elevated in both plasma and tumour tissues, while the opposite was the case for AEA, OEA and PEA [66]. In addition to these conflicting observations, the effects of these cannabinoids on cancer are quite obscure, as they have been shown to possess both potential antiproliferative and invasive effects [67,68].

These observable changes in endocannabinoids’ expression in PDAC cells demonstrate that there is a correlation between dysregulation of the eCBome system and pancreatic cancer. Comparing the extent of these changes between patients can help establish reliable prognostic markers for survival, therapy resistance and detection of PDAC [49].

## 5. Involvement of the Endocannabinoidome in Pancreatic Cancer

### 5.1. Cannabinoid Receptors

Cannabinoid CB_1_ and CB_2_ receptors play a part in pancreatic cancer through differing pathways. Promiscuity in these receptor-mediated effects is due to biased agonism of cannabinoid GPCR receptors, which can modulate their shape when bound to different endocannabinoid ligands, and effectuate multiple effects [69].

Many studies have shown that both CB_1_ and CB_2_ receptors mediate THC-mediated tumour growth inhibition [70,71,72]. CB_2_ receptor activation was found to exert the major anti-tumour effects in pancreatic cancer cell lines via an endoplasmic reticulum (ER) stress-regulated protein p8 dependent mechanism and activating transcription factor (ATF-4), leading to expression of the proapoptotic proteins CHOP and TRB3 [73]. The increase in antitumour genes is thought to occur via CB_2_ dependent de novo synthesis of ceramide [73,74]. Correspondingly, gemcitabine is thought to induce its antiproliferative effects by increasing reactive oxygen species (ROS) damage, disrupting ER homeostasis and increasing ER stress response [75]. This mechanism is consistent with the greater efficacy of gemcitabine in PDAC cells overexpressing CB_2_ receptors for this response [75]. This CB_2_-dependent proapoptotic mechanism can stimulate pharmacological interest as it does not exert the psychoactive effects of THC. Interesting to note was that low levels of THC induced a slight proliferative effect on PDAC [73]. This was achieved through non-cannabinoid receptors, suggesting that other potential eCBome receptors can promote tumour progression. Studies comparing endocannabinoids to THC did not show the biphasic effects of THC [73,76].

In contrast, the selective agonism of CB_1_ receptors by arachidonylcyclopropylamide (ACPA) and CB_2_ receptors by GW-405833, activated a common ROS mechanism that inhibited Panc1 cell proliferation and invasion, respectively [74]. Cannabinoid receptors stimulation was shown to cause a ROS-dependent increase in ATP/AMP ratio, inducing 5′ adenosine monophosphate-activated protein kinase (AMPK), inhibiting cancer cell growth, proliferation and invasion [74,77].

Furthermore, cannabinoid receptors are involved in decreasing the activation and migration of PSCs by pancreatic cancer cells in chronic pancreatitis [78]. Analysis of cannabinoids’ involvement with cells of the TME is important, as they can be a cause of the discrepancy between in vitro and in vivo pancreatic cancer experiments. Cannabinoid inhibition of inflammatory cytokines and extracellular matrix production is thought to be the mechanism behind this effect. Cannabinoid-dependent inhibition of PSCs migration has been associated with a decrease in matrix concentrations of metalloproteinase-2 (MMP-2), which is a marker for inflammation and fibrosis in chronic pancreatitis [78].

### 5.2. The TME and the Immune Endocannabinoid System

Pancreatic cancer has multiple known mechanisms that induce active systemic immunosuppression to decrease apoptotic and growth suppressing signals [79]. On top of this, due to the stromal reaction that forms the structure of PDAC tumours, immune cells have limited access to cancer cells [60]. Although the role of the eCBome in relation to cancer’s immune response is not fully understood, analysis of this relationship in PDAC is significant due to its notoriously poor response [80] along with the high expression of cannabinoids in immune cells [81].

Immune cells in the pancreatic TME are closely associated with the eCBome. Their relationship is referred to as the “immune endocannabinoid system” because immune cells express cannabinoid receptors (primarily CB_2_) and can release and metabolise endocannabinoid molecules [82]. Immune cell release of inflammatory cytokines, e.g., IL-6, TNF-α and IL-1β, is shown to increase CB_1_ and CB_2_ expression in cells of the TME [82,83]. Analysis of the immune response of the endocannabinoid ligands 2-AG, AEA and arachidonyl-2’-chloroethylamide, a highly selective cannabinoid CB1 receptor agonist, shows that the eCBome targets a variety of immune cell types including T cells, B cells NK cells, dendritic cells, macrophages, eosinophils, neutrophils, monocytes and mast cells, and results in numerous effects on tumours. These effects include induction of apoptosis, inhibition (AEA) or stimulation (2-AG) of migration, inhibition of cytokine production and control of degranulation via CB_1_ (Figure 3) [82]. B cells, as well as CD8^+^ and CD4^+^ T cells, are especially susceptible to the endocannabinoid influence listed above due to relatively greater expression of CB_1_ and CB_2_ receptors [82,84]. Presently, further research on the involvement of other eCBome lipids in PDAC, as well as clinical data to corroborate the results occurring with complex in vivo signalling, is urgently needed in order to translate this understanding into future therapies.

### 5.3. GPR55 Signalling

The immune TME appears to also be influenced by GPR55 receptor activation in colorectal cancer animal models, with observable differences between GPR55 knockout and wild-type mice [85]. The absence of GPR55 resulted in a lower concentration of the anti-tumour myeloid-derived tumour suppressing cells (MDSCs) [85]. Opposingly, the same knockout specimens possessed a greater amount of CD4^+^ and CD8^+^ cells, which are associated with longer survival [85]. Interestingly, 2-AG is able to increase MDSCs in PDAC without effects on T cells [84]. In an in vivo pancreatic cancer study on KPCG mice, which do not express GPR55 on PDAC cells, we confirmed that the absence of the GPR55 receptor significantly increased survival compared to the KPC control [60]. GPR55 was found to play a regulatory role in cell cycle progression, and its absence was shown to decrease cell proliferation and anchorage-dependent growth of PDAC cells. This occurred due to changes in the expression of cell-cycle regulatory cyclins, blocking the G1/S transition phase [60]. Interestingly, the tumour suppressor protein p53 was found to downregulate GPR55 in its anticancer effects [60].

In addition to biased agonism, varied effects of GPR55 inhibition could be due to some eCBome molecules acting as allosteric modulators instead of ligands of these receptors, allowing them to alter the effects of other cannabinoids [86].

GPR55 can also interact with other cannabinoid receptors, adding to the complexity of the eCBome signalling [87]. As is commonly observed in other GPCRs, GPR55 can oligomerise, in this case forming heteromers with CB_1_ and CB_2_ receptors as discussed below. Depending on the heteromer formation, different eCBome signalling changes related to cancer can be observed, including an enhancement of CB_1_ signalling with GPR55 [87], inhibition of GPR55 signalling [88], and modulation of CB_2_ activation [89].

Due to their ability to agonise GPR55 activity, LPI, specifically arachidonoyl-LPI, plays a role in pancreatic cancer progression [61]. Recently, we have shown that a reduction of GPR55 activation by LPI, caused by the inhibition of the LPI ABCC3 transporter, was able to decrease cancer cell growth and increase sensitivity to chemotherapy in prostate cancer [90].

### 5.4. Role of Endocannabinoid Receptor Heterodimers

In recent years, it has been shown that GPCRs can exist as multimers and, in particular, CB receptors are likely to form homodimers (same protein) or heterodimers (two different proteins) with other GPCRs or members of the EGFRs family [91].

Recent studies have in fact shown that, in breast cancer, CB_2_ forms heterodimers with epidermal growth factor receptor 2 (HER2), and treatment with THC exerts an anti-tumour effect in vitro and in vivo by disrupting CB_2_-HER2 dimers and inducing HER2 degradation [92,93]. In addition, GPR55 has been shown to form heterodimers with both CB_1_ and CB_2_ [94]. In human breast and prostate cancer cells, CB_2_ forms heterodimers with another GPCR, chemokine receptor CXCR4. CXCR4 is a key player in cancer progression and migration and targeting CB_2_R has been shown to reduce CXCR4-mediated mechanisms responsible for metastatic progression [95]. Interestingly, CXCR4 has been shown to be a key player in pancreatic cancer progression and metastasis [96]. We have recently observed that CB receptors, as well as HER2 and CXCR4, are overexpressed in pancreatic cancer cell lines, especially in a subset of cancer tumorspheres enriched in chemo-resistant cancer stem cells [97]. However, the existence of heterodimers involving members of the eCBome in PDAC has not been investigated.

Nevertheless, the ECS, and CB receptors heterodimers have the potential of becoming successful targets to develop novel therapeutic strategies to potentiate the effect of chemotherapy, in addition to the use of cannabinoids in palliative care.

## 6. Therapeutic Opportunities

### 6.1. Palliative Care

Currently, the use of medicinal cannabinoids remains restricted to palliative care. The synthetic cannabinoids nabilone and dronabinol are approved for the treatment of pain from pancreatic cancer symptoms and therapy, insomnia, and to stimulate appetite [98]. The mechanism of pain relief occurs through spinal, supraspinal and peripheral mechanisms, where CB_1_ receptors are involved with processing nociceptive signals and CB_2_ receptors can reduce inflammatory signals and release endogenous opioids [99,100].

### 6.2. Combined Chemotherapy

In addition to palliative treatment, evidence suggesting that the use of cannabinoids in combination with existing PDAC treatments can improve efficacy is growing, both in vivo and in vitro [17].

#### 6.2.1. In Vitro

Although tested extensively in vitro, little data is available on the effects of THC in combination with other PDAC chemotherapy treatments. The most promising example is observed with gemcitabine and other cannabinoids, which have synergistic cancer-inhibiting effects. Both selective CB_1_ (ACPA) and CB_2_ (GW-405833) receptor agonists were able to enhance the ROS-mediated inhibition of pancreatic cancer cell growth of gemcitabine [77]. Donadelli et al. noted that greater synergism occurred against gemcitabine-resistant cell lines, which is advantageous in combating the high chemotherapy resistance of PDAC [77]. Moreover, the potent synthetic cannabinoid agonist WIN55212,2 reduces migration of PSCs, which are also involved in chemoresistance, immunosuppression, cancer growth and invasion [78]. Concurrently, WIN55212,2 was shown to activate the ROS mediated ER stress pathway in pancreatic cancer cells but not healthy cells, suggesting adjuvant cannabinoid treatment can increase the cancer selectivity of chemotherapy cytotoxicity, allowing for lower required doses and fewer side effects [73]. Similarly, the synthetic GPR55 inhibitor 4-methoxy-1-naphthylfenoterol was able to enhance the cytotoxicity of gemcitabine and doxorubicin on PANC-1 cell lines and suggested that GPR55 inhibitors combat the multidrug resistance of PDAC [101].

#### 6.2.2. In Vivo

The phytocannabinoid cannabidiol (CBD) has also proven to enhance PDAC treatment with gemcitabine in vivo. KPC mice treated with CBD and gemcitabine were found to have an almost threefold survival rate compared to gemcitabine alone [60]. As a GPR55 antagonist, CBD is thought to both inhibit cell cycle progression and potentiate the effects of gemcitabine through GPR55 receptor inhibition [60]. In addition to GPR55 inactivation, CBD may effectuate additional anti-cancer effects through other targets in the eCBome, including negative allosteric modulation of CB_1_ receptors and partial agonism of CB_2_ receptors [102]. There is no clinically significant data for the use CBD/gemcitabine adjuvant therapy in humans at the moment, however, a small cohort study on nine patients reported a twofold increase in survival compared to the average [103].

### 6.3. Cannabinoids in Combined Radiotherapy

In addition to chemotherapy, pancreatic radiotherapy studies have shown that cannabinoids have synergistic effects with radiotherapy treatment. Yasmin-Karim et al. found CBD enhanced radiotherapy tumour inhibition in in vitro pancreatic cancer experiments. Additionally, their study also implemented smart biomaterial delivery of cannabinoids to PDAC cancer cells. These biomaterials change their structure and release CBD in response to stimuli like pH, the tumour microenvironment or radiation intensity, allowing for selective delivery of CBD to cancer cells for a prolonged duration. When tested on mice injected with PANC-02 cells, smart biomaterials containing CBD in conjunction with radiotherapy resulted in a statistically significant increase in survival [104].

### 6.4. Cannabinoids as Immunotherapy Agents

As the ECS in immune cells is known to play a role in pancreatic cancer progression, cannabinoid treatment alone or as an adjuvant to immunotherapy is a valuable pharmacological approach to explore. Sadly, the efficacy of endocannabinoids for this use is still not clear. THC has been shown to act as an immunosuppressor and even enhance cancer growth in some cancers [105]. On the other hand, in pancreatic cancer cells and mice models, both THC and CBD have recently been tested with positive results. These phytocannabinoids inhibited the proliferation of PDAC cells and PSCs, as well as reduced the PSCs-mediated activation of PDAC cells [106]. CBD and THC acted via a p-21 activated kinase 1 (PAK1) dependent mechanism to decrease PD-L1, which is thought to inhibit pancreatic cancer growth via immune checkpoint blockade [106].

Additionally, although 2-AG administration was observed to promote tumour suppressive effects, it overall managed to inhibit pancreatic cancer cell proliferation in vitro [84]. These ambiguous results could suggest that poor understanding of the eCBome system’s effects, which are likely to involve multiple regulatory endocannabinoid mechanisms, are preventing our comprehension of molecular mechanisms and in vivo results.

### 6.5. Inhibitors of Endocannabinoid-Metabolising Enzymes

The use of molecules interfering with endocannabinoid-metabolising enzymes represents an intriguing additional therapeutic strategy. A recent review summarised the relevant preclinical studies with FAAH and MAGL inhibitors in cancer [107]. This field of investigation is still in its infancy and more studies are required to explore fully the potential of interfering with enzymes responsible for endocannabinoids’ metabolism in cancer settings.

## 7. Conclusions

Increasing evidence suggests that targeting the eCBome in PDAC offers interesting opportunities for future therapies. Despite the accumulating substantiation of the role played by eCBome members in PDAC and, consequently, the potential to use pharmacological tools that interfere with eCBome-related pathways, there are very few clinical studies to support these claims. Our search of available databases (ClinicalTrails.gov, accessed on 1 February 2022 ) for ongoing clinical studies related to pancreatic cancer and the eCBome reveals a limited number of current trials (Table 1). However, the increased interest in the use of cannabinoid-related drugs and the increased number of studies in this field might predict a better future for eCBome treatments in PDAC.

## Figures and Tables

**Figure 1 biomolecules-12-00320-f001:**
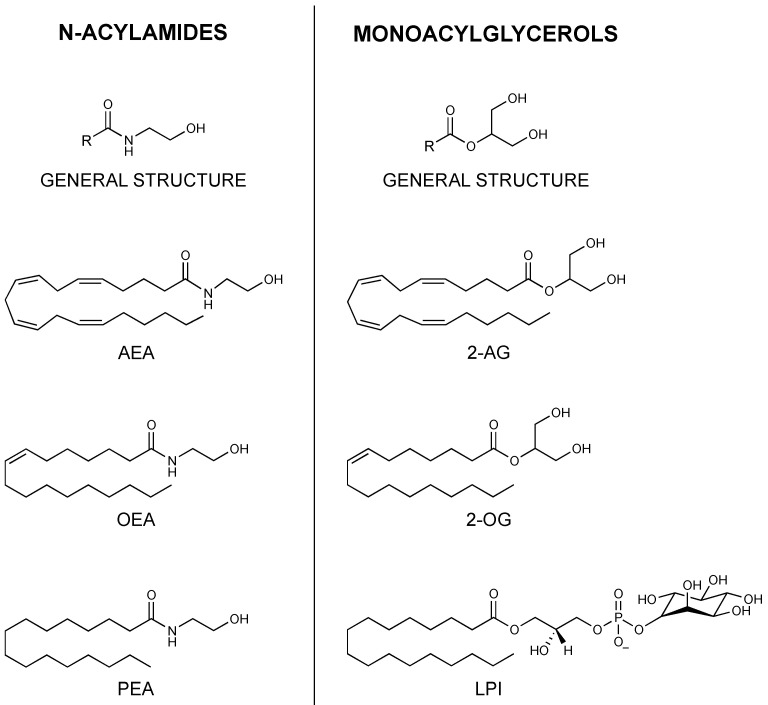
General structure and examples of the two principal eCBome ligand families: *N*-acylamides and monoacylglycerols. Anandamide, AEA; 2-arachidonoylglycerol, 2-AG; oleoylethanolamide, OEA; 2-oleoylglycerol, 2-OG; palmitoylethanolamide, PEA; lysophosphatidylinositol, LPI.

**Figure 2 biomolecules-12-00320-f002:**
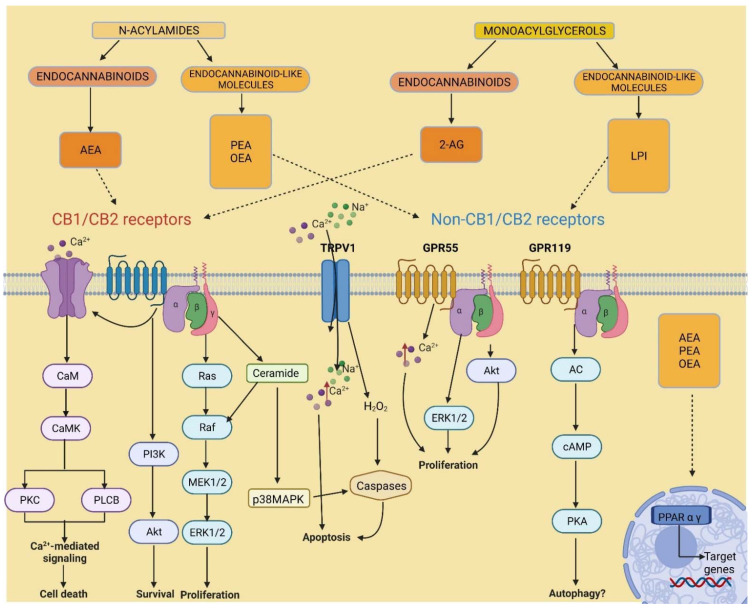
Endocannabinoidome signalling. Adenylyl cyclase, AC; anandamide, AEA; calmodulin, CaM; calmodulin-dependent protein kinase, CaMK; cyclic adenosine monophosphate, cAMP; cannabinoid receptor 1, CB1; cannabinoid receptor 2, CB2; extracellular-regulated protein kinase, ERK; G protein-coupled receptor 55, GPR55; G protein-coupled receptor 119, GPR119; lysophosphatidylinositol, LPI; mitogen-activated protein kinase, MEK; p38 mitogen-activated protein kinase, p38MAPK; oleoylethanolamide, OEA; palmitoylethanolamide, PEA; peroxisome proliferator-activated receptor, PPAR; phosphoinositide 3-kinase, PI3K; phospholipase C beta, PLCB; protein kinase A, PKA; protein kinase B, Akt; protein kinase C, PKC; rapid accelerated fibrosarcoma, Raf; rat sarcoma, Ras; transient receptor potential of the vanilloid type-1, TRPV1.

**Figure 3 biomolecules-12-00320-f003:**
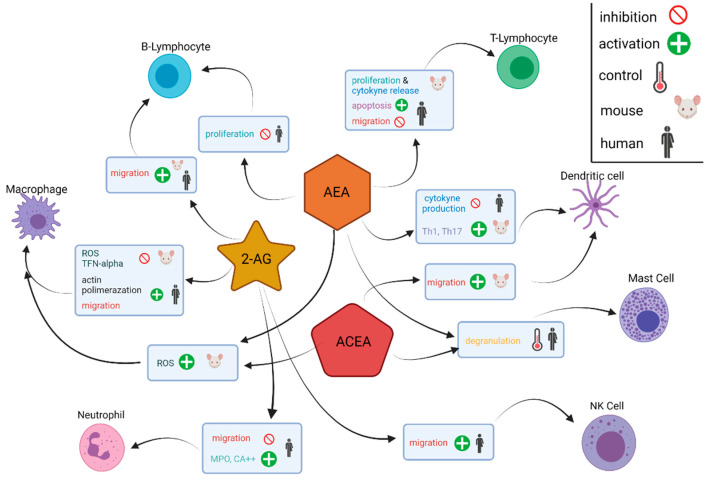
Diagram depiction of the molecular effects of the cannabinoids AEA, ACEA and 2-AG on different immune cells. Immune cells within the tumour microenvironment express receptors of the endocannabinoidome and release endocannabinoids and endocannabinoid-like molecules. Arachidonyl-2’-chloroethylamide, ACEA; anandamide, AEA; 2-arachidonoylglycerol, 2-AG; myeloperoxidase, MPO; reactive oxygen species, ROS; T helper 1 cells, Th1; T helper 17 cells, Th17; tumour necrosis factor-alpha, TNF-alpha.

**Table 1 biomolecules-12-00320-t001:** Selected ongoing studies on endocannabinoidome targets of therapy in pancreatic cancer.

Disease	Title	Reference	Location	Status
Pancreatic Neoplasm, Cachexia	The effect of Cannabis in Pancreatic Cancer	NCT03245658	Naestved, Denmark	Unknown
Pancreatic Cancer Non-resectable and Metastatic, Chemotherapy-induced Nausea and Vomiting	Efficacy and safety of Dronabinol in the improvement of Chemotherapy-induced and Tumour-related Symptoms in Advanced Pancreatic Cancer	NCT03984214	Oberösterreich, Klagenfurt, Leoben, Salzburg, St. Veit an der Glan, Steyr, Vienna, Zams Austria	Recruiting
Pancreatic Cancer	Nutrition and Pharmacological Algorithm for Oncology patients’ study	NCT04155008	New York, United States	Recruiting
Hepatocellular Carcinoma, Pancreatic Cancer	A Study of Dexanabinol in Combination with Chemotherapy in Patients with Advanced Tumours	NCT02423239	Germany, Poland, Spain, United Kingdom	Unknown
Cancer of Pancreas, Liver, Rectum, Colon, Gall Bladder	A Study of the Efficacy of Cannabidiol Patients with Multiple Myeloma, Glioblastoma Multiforme, and GI Malignancies	NCT03607643	Florida, United States	Unknown
Hepatocellular Carcinoma, Cholangiocarcinoma, GastroEsophageal, Colorectal and Pancreatic cancer	TPST-1120 as Monotherapy and in Combination with Nivolumab in Subjects with Advanced Cancers	NCT03829436	California, Florida, Maryland, Michigan, New York, North Carolina, Oklahoma, Pennsylvania, Tennessee, United States	Recruiting

## Data Availability

Databases MEDLINE/PubMed, Google Scholar and EMBASE were searched for studies on the endocannabinoidome pathway, cannabinoids, and pancreatic cancer. The search terms ‘endocannabinoidome’, ‘cannabinoids’, ‘pancreatic cancer’, ‘regulation’ and ‘therapy’ were used. Publications from the reference lists of recovered articles were also reviewed. Articles were published between 1967 and 2022. Articles not using the English language and non-peer-reviewed were excluded. The final database search and ClinicalTrials.gov search was performed on 1 February 2022.

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
