# Peer review of "Targeting the Endocannabinoidome in Pancreatic Cancer"

_biomolecules, 2022, doi:10.3390/biom12020320_

Round 1
Reviewer 1 Report
In this manuscript titled “Targeting the Endocannabinoidome in Pancreatic Cancer,” the authors give an overview of the endocannabinoidome (eCBome) components and their role in various aspects of cancer. While new targeting strategies to improve therapeutic outcome in pancreatic cancer is direly awaited and, as such, is a topic of great importance, I have some reservations about the manuscript as is.
Major comments:
1, I would suggest thorough proofreading, as minor typographical and grammatical errors were seen throughout the manuscript.
2, Authors should take care to cite suitable references more often: for a non-trivial claim, a reference is requested. Of note, if often seemed that references were collectively cited after several sentences in the main text, which makes it hard for readers (especially those new to the field) to decipher form which reference exactly the information being presented emanated.
3, I surmise that most readers are not so familiar with the concept of eCBome and its various components. I believe the addition of a figure depicting the various components (e.g., ligands/receptors/effectors), including the differing signaling capabilities of the various endogenous/synthetic ligands, would help the readers’ comprehension.
4, Please make sure whether a particular claim being made in the manuscript about the eCBome is based on data on pancreatic cancer or not, as it was sometimes ambiguous.
5, Figure 2: Please make sure the information depicted in the figure is accompanied by sufficient explanation within the main text or the figure legend. Some information seems to only appear in this figure.
Minor comments:
1, Line 213-214: Regarding the GPR119 analyses on the Human Protein Atlas data, if this is a published observation, please provide a suitable reference. If not, make sure it is clear that this is an unpublished observation (and I believe the HPA requires the authors to cite particular publications when using their data for analysis).
2, Table 1: The authors should clarify when they performed the search on ClinicalTrials.gov.
Author Response
In this manuscript titled “Targeting the Endocannabinoidome in Pancreatic Cancer,” the authors give an overview of the endocannabinoidome (eCBome) components and their role in various aspects of cancer. While new targeting strategies to improve therapeutic outcome in pancreatic cancer is direly awaited and, as such, is a topic of great importance, I have some reservations about the manuscript as is.
We would like to take this opportunity to thank the Reviewer for his/her comments that helped improving our manuscript. We have now revised our manuscript as requested to take into account the Reviewer’s comments (as described in details below)
Major comments:
1, I would suggest thorough proofreading, as minor typographical and grammatical errors were seen throughout the manuscript.
A thorough proofreading has been performed
2, Authors should take care to cite suitable references more often: for a non-trivial claim, a reference is requested. Of note, if often seemed that references were collectively cited after several sentences in the main text, which makes it hard for readers (especially those new to the field) to decipher form which reference exactly the information being presented emanated.
We have added more references as suggested and avoided, when possible, collective referencing.
3, I surmise that most readers are not so familiar with the concept of eCBome and its various components. I believe the addition of a figure depicting the various components (e.g., ligands/receptors/effectors), including the differing signaling capabilities of the various endogenous/synthetic ligands, would help the readers’ comprehension.
As suggested we added a figure (Figure 2) depicting the various components including the differing signalling capabilities of the various ligands
4, Please make sure whether a particular claim being made in the manuscript about the eCBome is based on data on pancreatic cancer or not, as it was sometimes ambiguous.
We assessed all claims and specified if relative to pancreatic cancer or to other conditions
5, Figure 2: Please make sure the information depicted in the figure is accompanied by sufficient explanation within the main text or the figure legend. Some information seems to only appear in this figure.
Explanation provided
Minor comments:
1, Line 213-214: Regarding the GPR119 analyses on the Human Protein Atlas data, if this is a published observation, please provide a suitable reference. If not, make sure it is clear that this is an unpublished observation (and I believe the HPA requires the authors to cite particular publications when using their data for analysis).
Reference added
2, Table 1: The authors should clarify when they performed the search on ClinicalTrials.gov.
Details on data search have been included in the Data availability Statement section
Reviewer 2 Report
This is an intersting and well written review regarding the role of cannabinoids in the treatment of PC.
Major points:
I recommend to present more information on the sources and types of cannabinoids in the introduction.
In addition, methods of the literature searching should be presented (databases, key words, selection criteria as well as a count and types of selected articles.
The articles regarding therapeutic opportunities should be divided into: in vitro and in vivo papers. I recommend to summarize presented in vitro and in vivo article in the table.
Minor points:
The references should be presented according to the guidelines for authors.
Author Response
This is an interesting and well written review regarding the role of cannabinoids in the treatment of PC.
We would like to take this opportunity to thank the Reviewer for his/her comments that helped improving our manuscript. We have now revised our manuscript as requested to take into account the Reviewer’s comments (as described in details below)
Major points:
I recommend to present more information on the sources and types of cannabinoids in the introduction.
As suggested we have included more information on the sources and types of cannabinoids in the introduction.
In addition, methods of the literature searching should be presented (databases, key words, selection criteria as well as a count and types of selected articles.
Details on data search have been included in the Data availability Statement section
The articles regarding therapeutic opportunities should be divided into: in vitro and in vivo papers. I recommend to summarize presented in vitro and in vivo article in the table.
We have divided the chapter into in vitro and in vivo papers
Minor points:
The references should be presented according to the guidelines for authors
References formatting performed
Round 2
Reviewer 1 Report
Thank you for the revisions. I felt that the manuscript was greatly improved.
A very minor comment: please proofread the legend for Figure 2.
-"cyclic CAmp" probably should be cyclic AMP, cAMP.
-Erk seems misplaced.
-Transient is misspelled for TRPV1.
Author Response
We would like to thank the Reviewer for his/her comments that helped improving our manuscript. We have now revised Figure 2 and Figure 2 legend as requested to take into account the Reviewer’s comments
Reviewer 2 Report
The authors have improved their manuscript according to my suggestions. I recommend it for publication.
Author Response
We would like to thank the Reviewer for his/her comments that helped improving our manuscript.